# MagicVFX: Visual Effects Synthesis in Just Minutes

Jiaqi Guo
University of Electronic Science and
Technology of China
Chengdu, China
guoojq@gmail.com

Lianli Gao*
Shenzhen Institute for Advanced
Study, University of Electronic
Science and Technology of China
Shenzhen, China
lianli.gao@uestc.edu.cn

Junchen Zhu
University of Electronic Science and
Technology of China
Chengdu, China
junchen.zhu@hotmail.com

Jiaxin Zhang
University of Electronic Science and
Technology of China
Chengdu, China
aria.jiaxinz@gmail.com

Siyang Li
University of Electronic Science and
Technology of China
Chengdu, China
simonhas3cats@gmail.com

Jingkuan Song
Shenzhen Institute for Advanced
Study, University of Electronic
Science and Technology of China
Shenzhen, China
jingkuan.song@gmail.com

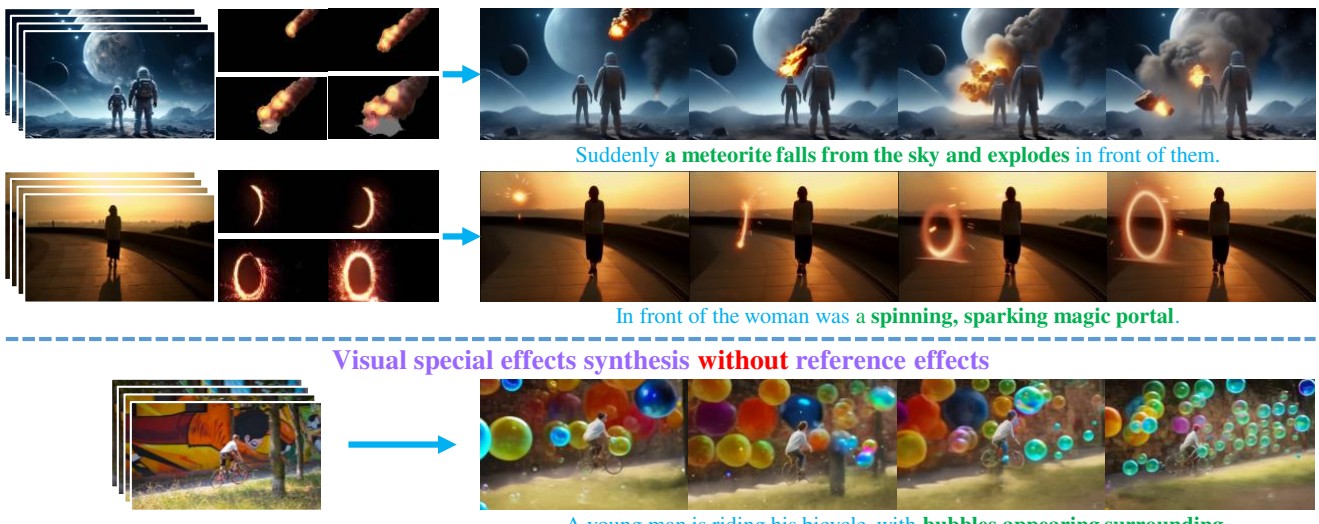

**Visual special effects synthesis with reference effects**

Suddenly **a meteorite falls from the sky and explodes** in front of them.

In front of the woman was a **spinning, sparking magic portal**.

**Visual special effects synthesis without reference effects**

A young man is riding his bicycle, with **bubbles appearing surrounding**.

Drift car performance on a turn of the race track, emitting a **magical light**.

**Figure 1: Two paradigms of AIGC-based visual effects synthesis: synthesis with reference effects(SRE) and synthesis without reference effects(SNRE). Both paradigms take a base video and a textual description as input, and the paradigm with reference effects additionally has a visual effects video as input. Top: Our results for the paradigm with reference effects. Bottom: Our results for the paradigm without reference effects.**

*Corresponding author.

## Abstract

Visual effects synthesis is crucial in the film and television industry, which aims at enhancing raw footage with virtual elements for

© 2024 Copyright held by the owner/author(s). Publication rights licensed to ACM.
ACM ISBN 979-8-4007-0686-8/24/10
https://doi.org/10.1145/3664647.3681516

greater expressiveness. As the demand for detailed and realistic effects escalates in modern production, professionals are compelled to allocate substantial time and resources to this endeavor. Thus, there is an urgent need to explore more convenient and less resource-intensive methods, such as incorporating the burgeoning Artificial Intelligence Generated Content (AIGC) technology. However, research into this potential integration has yet to be conducted. As the first work to establish a connection between visual effects synthesis and AIGC technology, we start by carefully setting up two paradigms according to the need for pre-produced effects or not: synthesis with reference effects and synthesis without reference effects. Following this, we compile a dataset by processing a collection of effects videos and scene videos, which contains a wide variety of effect categories and scenarios, adequately covering the common effects seen in films and television industry. Furthermore, we explore the capabilities of a pre-trained text-to-video model to synthesize visual effects within these two paradigms. The experimental results demonstrate that the pipeline we established can effectively produce impressive visual effects synthesis outcomes, thereby evidencing the significant potential of existing AIGC technology for application in visual effects synthesis tasks. Our dataset can be found in https://github.com/ruffiann/MagicVFX.

## CCS Concepts

• **Computing methodologies → Computer vision**.

## Keywords

visual effects, video synthesis, diffusion models

**ACM Reference Format:**
Jiaqi Guo, Lianli Gao, Junchen Zhu, Jiaxin Zhang, Siyang Li, and Jingkuan Song. 2024. MagicVFX: Visual Effects Synthesis in Just Minutes. In *Proceedings of the 32nd ACM International Conference on Multimedia (MM '24), October 28-November 1, 2024, Melbourne, VIC, Australia.* ACM, New York, NY, USA, 9 pages. https://doi.org/10.1145/3664647.3681516

## 1 Introduction

Visual effects are indispensable components of films, television, and even social media videos. Detailed and realistic visual effects often provide audiences with the ultimate visual pleasure. For example, in disaster films, the tsunamis or fires processed with visual effects will be more immersive for the audience, enabling audiences to feel as though they are part of the scene.

As the film and television industry advances, audience expectations for visual effects have increased dramatically, covering both realistic phenomena that are challenging to capture (such as explosions and car accidents) and fantastical effects that do not exist in reality (such as magical shields and teleportation circles). Meeting these expectations requires significant investment in high-performance hardware and professional talent, making the process costly and complex. Therefore, there is an immediate need to improve current approaches and workflows of visual effects synthesis. In light of this, we propose leveraging the rapidly evolving Artificial Intelligence Generated Content (AIGC) technology for visual effects synthesis, aiming to improve the efficiency in the current film and television production industry. To the best of our knowledge, we

are the first to attempting to bridge these two areas, which means we will be exploring the feasibility of this idea from scratch.

First, we relate AIGC to visual effects synthesis and propose the task definition. In film and television production, creating visual effects often involves capturing a base video and then adding designated effects to specific locations or objects within it. These effects generally fall into two categories: those that do not require customization and can be described using natural language, such as flames and lightning, and those that require customization, such as the magical shields seen in Doctor Strange. The latter are typically pre-produced by professionals as samples before being integrated into the scene. Consequently, we empirically divide AIGC-based visual effects synthesis into two paradigms: synthesis with reference effects (SRE) and synthesis without reference effects (SNRE), as shown in Figure 1. Both paradigms utilize a base video as inputs, with SRE additionally requires a customized visual effects video.

Secondly, we compile a dataset for evaluation purposes. Our data acquisition approach is shown in Figure 3. This dataset comprises two types of videos: pre-produced effects videos and base videos. The effects videos are collected from online sources, with each effect being manually annotated with a brief description. For the base videos, we select clips appropriate for effects augmentation from both online sources and public video dataset [24], each accompanied by detailed textual description crafted manually. Utilizing these two distinct video types as a foundation, we further processed them to create samples tailored for both SRE and SNRE paradigms. Our dataset encompasses a wide range of visual effects commonly encountered in the film and television industry, enabling a comprehensive assessment of the capabilities of synthesis methods.

In the third phase, utilizing a pre-trained text-to-video generator, VideoCrafter2 [6], we design a simple yet effective pipeline in Figure 4 capable of achieving visual effects synthesis under both paradigms. Specifically, users are required to designate an area for effects on a base video using a mask. Should a reference effect be available, it is pasted to the marked area, creating an initial video; if not, the original base video remains unchanged. Next, the initial video undergoes the addition of low-level noise, leading to a process of denoising to generate the final output. Throughout this procedure, we enhance the effects synthesis by using descriptions without effects as negative prompts and employing user masks to replace cross-attention maps. The experimental results validate the effectiveness of our pipeline in accomplishing visual effects synthesis, also highlighting the research potential of AIGC-based visual effects synthesis tasks.

In summary, our contributions are as follows:

- To the best of our knowledge, we are the first to introduce AIGC techniques to visual effect synthesis which only takes few minutes to generate complex and high-quality special effects in videos.
- We define the visual effect synthesis task, and correspondingly collect a basic dataset encompassing common visual effects for assessing the ability of visual effects synthesis methods.
- We build a training-free base model with extensive experiments, and demonstrate the potential of the pretrained video diffusion model in visual effect synthesis.

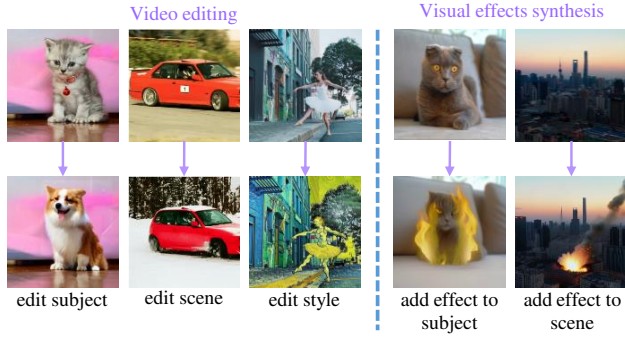

**Figure 2: Differences between visual effects synthesis tasks and video editing tasks. Visual effects synthesis aims to add visual content to the original input, while video editing tasks are concerned with changing some part of the visual content in the original input.**

## 2 Related work

In the realm of visual effects, traditional methodologies have predominantly relied on software-based approaches [27] and conventional techniques [22]. These methods only optimize the software at a certain step and do not improve the overall workflow. In contrast, our work pioneers the application of AIGC technology to the synthesis of visual effects, revolutionizing the workflow and making it possible to synthesize visual effects in just minutes.

**Video Composition.** In the realm of video composition, recent advancements have primarily concentrated on achieving temporal consistency [7] and seamless blending [12, 16, 21, 28]. But video composition concentrates on real entities such as humans and animals, neglecting the compositing of virtual objects such as visual effects.

**Text-driven Video Editing.** Text-driven video editing [4, 8, 11, 17–19, 25, 26, 31, 32] involves modifying the video conditional on the text, which is similar to our proposed paradigm of synthesis without reference effects. However, as illustrated in Figure 2, the focus of these two tasks is different. The goal of visual effects synthesis is to enhance the original input with additional visual elements, whereas video editing tasks [10, 17, 20, 32] focus on modifying certain aspects of the visual content in the original input.

**Text-to-video Models.** Advancements in diffusion models have led to breakthroughs in text-to-video (T2V) generation [5, 33, 35] and multi-modal models [1–3, 9]. Several startups, like Pika Labs[1], Moonvalley[2], and Genmo[3], have released impressive text-to-video generation services. However, these rely on proprietary datasets and models, hindering broader research and development. Among open-source models, some [5, 13–15, 29] often yield videos of lower resolution. While I2VGen-XL [34] can generate high-resolution outputs, it falls short in text-to-video tasks, which are essential to our study. Therefore, we have selected VideoCrafter2 [6] for our work, which is the SOTA open-source text-to-video generator.

---
[1]https://www.pikalabs.com/
[2]https://moonvalley.ai/
[3]https://www.genmo.ai/

## 3 Problem formulation and datasets

### 3.1 Problem Formulation

In this study, we explore integrating AIGC technology with synthesizing visual effects, aiming to augment the production productivity. Following an in-depth investigation of the industrial workflows and techniques utilized in creating visual effects within the cinematic sphere, we propose categorizing AIGC-based visual effects synthesis into two distinct paradigms: synthesis with reference videos (SRE) and synthesis without reference videos (SNRE). These two paradigms validly cover the spectrum of synthesis scenarios encountered in practical applications.

**Synthesis with Reference Effects (SRE).** In the film industry, fantasy and science fiction genres command a significant market share, featuring an abundance of scenes rich in imaginative special effects. Such movies often delve into concepts like interstellar travel, time travel, and magical battles. These scenarios necessitate a wide array of custom special effects, meticulously pre-designed by visual effects designers and seamlessly integrated during post-production to bring these visionary concepts to life on screen. We describe this type of need for pre-produced effects as synthesis with reference effects (SRE). SRE process can be formulated as:

$$V_o = \text{SRE}(V_{base}, V_{ref}, M_{user}) \tag{1}$$

where $V_{base}$ is the base video, $V_{ref}$ is the reference effect video, and $M_{user}$ represents the designated location for effect integration. The output $V_o$ is the composite video with the reference effect seamlessly and harmoniously integrated.

**Synthesis without Reference Effects (SNRE).** Beyond the imaginative and bespoke effects, there exists a spectrum of common effects primarily crafted to simulate scenarios challenging to capture in the real world. These include weather phenomena, natural disasters, and smoke, as well as effects related to water and fire dynamics. Such effects do not require pre-production design but can be directly adapted to fit within scenes. We refer to the synthesis of these types of effects as synthesis without reference effects (SNRE). SNRE can be represented by:

$$V_o = \text{SNRE}(V_{base}, T, M_{user}) \tag{2}$$

where $T$ should be a textual description that is sufficiently descriptive of the effect, or a description of the expected output. $V_o$, $V_{base}$ and $M_{user}$ have the same meaning as in SRE.

After formally defining these two paradigms, we can use the AIGC model to fit and solve the problem, thus meeting the need for efficiency and automation in the visual effects field and speeding up the production process.

### 3.2 Dataset Construction

Since AIGC-based visual effects synthesis is a novel work, there is no publicly available dataset that evaluates how well the method performs on this task. Hence, we collect and construct a dataset named **VFX-307** to fill this gap. Our data acquisition method is shown in Figure 3.

**Data Collection.** VFX-307 contains two types of data, base videos and effect videos. We gather an assortment of base videos from two sources: open-source video datasets (e.g. DAVIS-2017 [24]) and online sources. We meticulously curate scenes suitable for

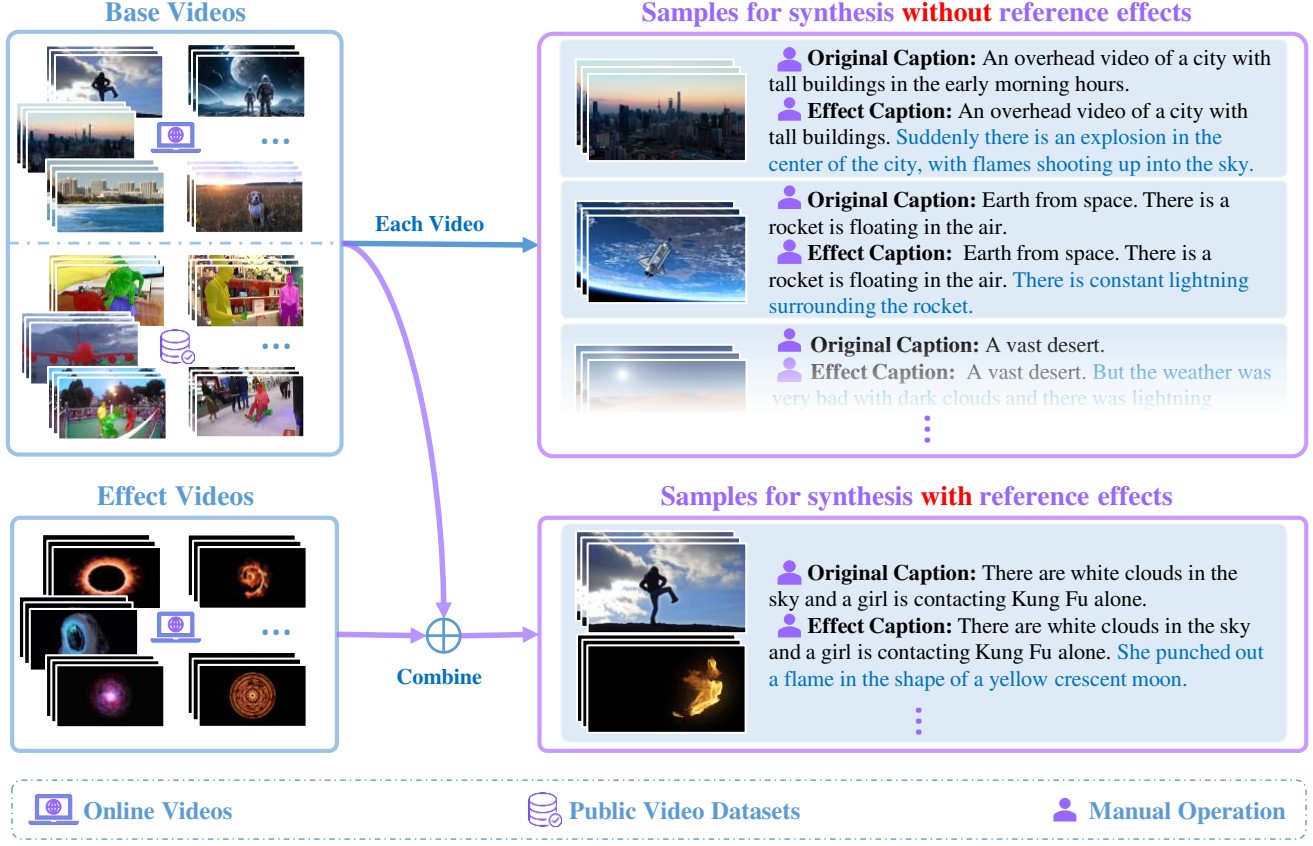

**Figure 3: Our data acquisition method. We first collect a series of effects videos and base videos. Then we manually process them to obtain samples suitable for SRE and SNRE, respectively.**

adding visual effects, prioritizing those with stable camera motion, clear settings, and distinct actions. We manually trim the videos to retain only segments shorter than 300 frames and standardize the resolution to 1080$p$ with a 16 : 9 aspect ratio. This process resulted in resulted in a collection of 117 base videos. For effect videos, we compile 190 pre-produced standalone effects videos uploaded by professionals on public networks. Despite varying resolution and duration, they all have a 16 : 9 aspect ratio. This compilation contains a wide variety of effect categories, adequately covering the common effects seen in films and the TV industry. Upon collecting the two types of videos, we manually provide each base video with a detailed textual description as the original caption and each effect video with a brief description.

**Samples Creation.** Based on these two categories of videos, we further process them to create sample sets for SRE and SNRE. Specifically, we manually pair each base video with an effect video to form valid and logical combinations. For each combination, we merge the original caption of the base video with the brief description of the effect video to generate a new effect caption. This results in a sample in the form of a quadruplet comprising the original caption, the effect caption, the base video, and the effect video. We created 75 such samples for SRE. For SNRE, we manually add a

special effect description to the original caption of each base video to formulate an effect caption. This yields a triplet consisting of the original caption, the effect caption, and the base video. Each base video is associated with 1 to 4 such triplets, culminating in a total of 175 samples.

## 4 Method

In this section, we explore to implement the task of visual effects synthesis using a pre-trained text-to-video generation model. We first give an overview of our synthesis pipeline in Section 4.1 followed by detailing the implementation of utilizing this pipeline for SRE and SNRE, in Section 4.2 and 4.3, respectively.

### 4.1 Overview

Our pipeline offers a straightforward yet potent approach to synthesizing visual effects, as shown in Figure 4. First, users are required to define a mask $M_{user}$ to specify where effects should be applied to a base video $V_{base}$. In SRE, where a reference effect $V_{ref}$ is available, it is pasted onto the designated area to create an initial video $V_{init}$; in SNRE, the original $V_{base}$ is utilized directly as $V_{init}$. Following

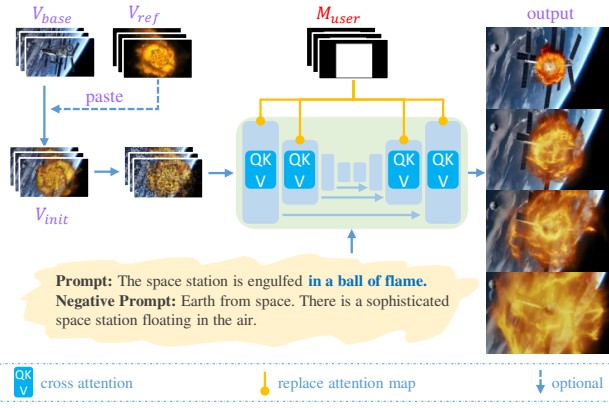

**Figure 4: Our proposed pipeline for visual effect synthesis. For SRE, we first paste $V_{ref}$ on $V_{base}$ to form $V_{init}$, while $V_{base}$ is used directly as $V_{init}$ for SNRE. Then, low-level noise is introduced into $V_{init}$, followed by a denoising generation process. During this process, we use the original caption as negative prompt and employ $M_{user}$ replacing cross-attention maps in specific UNet layers to enhance the synthesis.**

this, low-level noise is added to the $V_{init}$, which then undergoes a process of denoising to generate the final output.

Throughout the SNRE procedure, we employ two key techniques to enhance effect synthesis: 1. utilizing original captions of the base videos as negative prompts to ensure the generated results do not remain unchanged. 2. utilizing user masks as cross-attention maps in specific UNet layers, thereby strengthening the connection between the effect text token and its intended location.

Next, we detail the design of our pipeline for both paradigms and provide an in-depth explanation of our key techniques.

### 4.2 Synthesis with reference effects

In the scenario of visual effects synthesis with reference effects, our goal is to reasonably integrate a pre-produced effect video $V_{ref}$ into a specific location within the base video $V_{base}$. Professional engineers often adjust parameters like color and contrast to harmonize the overall visual effect and may alter elements in the original footage to ensure the composite video's coherent. For example, when facing destructive effects like explosions, it may be necessary not only to adjust the effects themselves but also to modify elements within the original scene to depict the destruction.

This complex and time-consuming manual process can be greatly simplified by employing a pre-trained video generation model with advanced semantic understanding and generalization capabilities. [10, 12, 23] showed that introducing an appropriate level of noise into the input image, and using this noisy image as the starting point for a denoising generation process, can effectively make the output more harmonious while preserving the structural information of the input.

Inspired by this finding, we first paste the reference effect video $V_{ref}$, at a certain value of transparency discussed in Section 5.2, onto the designated location by $M_{user}$ of the base video $V_{base}$ to

create an initial composite video $V_{init}$. Then, we add low-level noise to $V_{init}$ to serve as a starting point for generation. During the denoising generation process, we use the textual description of the scene containing the special effect as a conditional guidance. Unlike methods like [23, 30], which generate realistic images and videos by adding noise to sketches, our purpose in adding noise is not to provide a rough prior, but to facilitate adding effects.

### 4.3 Sythesis without reference effects

In the scenario of synthesis without reference effects, we rely on only textual descriptions to guide the model in generating the desired effects. First, we attempt the same method utilized for SRE, which involves directly using the base video as the initial video, introducing low-level noise to establish a generation starting point, and then guiding the generation process with text description containing the desired effects.

However, lacking the information provided by reference effects, the model suffers from two problems: the inability to produce visible effects and the inability to synthesis the effects to the correct location. Therefore, we treat this method as a baseline and employ the following two techniques to improve our approach in order to solve these problems.

**Original Captions as Negative Prompts.** In conditional diffusion model with classifier-free guidance, the model typically predicts noises $\epsilon^t$ and $\epsilon_{neg}^t$ at each step $t$ conditioned separately on positive and negative prompts. These two noises are then linearly combined to produce the final noise $\hat{\epsilon}_t$ for the current timestep $t$:

$$\hat{\epsilon}_t = \epsilon_t^{neg} + \omega * (\epsilon_t - \epsilon_t^{neg}), \tag{3}$$

where $\omega$ is the classifier-free guidance scale. This approach guides the model to avoid generating elements or themes mentioned in the negative prompts within the output image. Here, to emphasize visual effects in the output video, our method uses original textual descriptions of the base video that do not contain effects text directly as negative prompts.

**User Masks as Attention Maps.** As illustrated in subfigure (b) of Figure 9, the result of synthesizing 'flames' effect guided only by text description. The desired flames do not appear on the car as expected, but on the grass next to it. We deduce that this issue stemmed from the model's inability to adequately focus the attention of the token 'flames' on the location we expected. Thus, we force a connection between the the 'flames' and the location we expected by introducing a user mask and replacing the cross-attention map in the UNet. And as discussed in Section 5.2, we modify the value of non-zero pixels in the mask and the action position of the mask to get better results.

## 5 Experiments

In this section, we first show the results of our method in Section 5.1 for both SRE and SNRE paradigms. Subsequently, we report a series of ablation experiments in Section 5.2 to demonstrate the reasons for some of the design choices in our method.

### 5.1 Performance Evaluation

**Baseline.** Since we are the first work to explore AIGC-based visual effects synthesis, there is no applicable methodology for us

base video        effect video        output

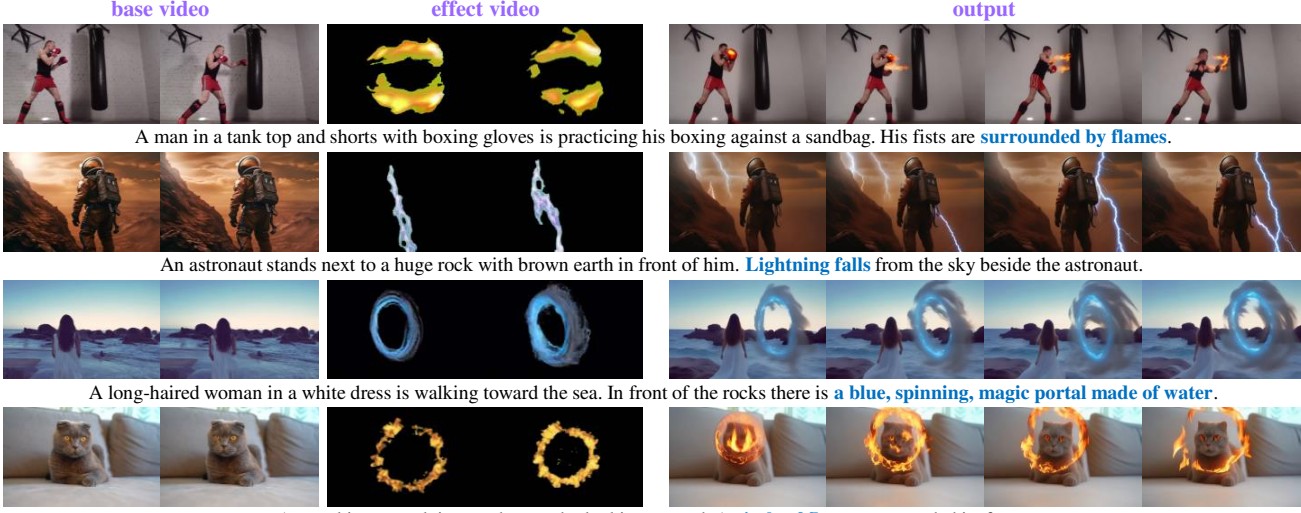

A man in a tank top and shorts with boxing gloves is practicing his boxing against a sandbag. His fists are **surrounded by flames**.

An astronaut stands next to a huge rock with brown earth in front of him. **Lightning falls** from the sky beside the astronaut.

A long-haired woman in a white dress is walking toward the sea. In front of the rocks there is **a blue, spinning, magic portal made of water**.

A gray kitten was lying on the couch, looking around. A **circle of flames** surrounded its face.

**Figure 5: Qualitative results for synthesis with reference effects (SRE).**

base video        baseline        ours

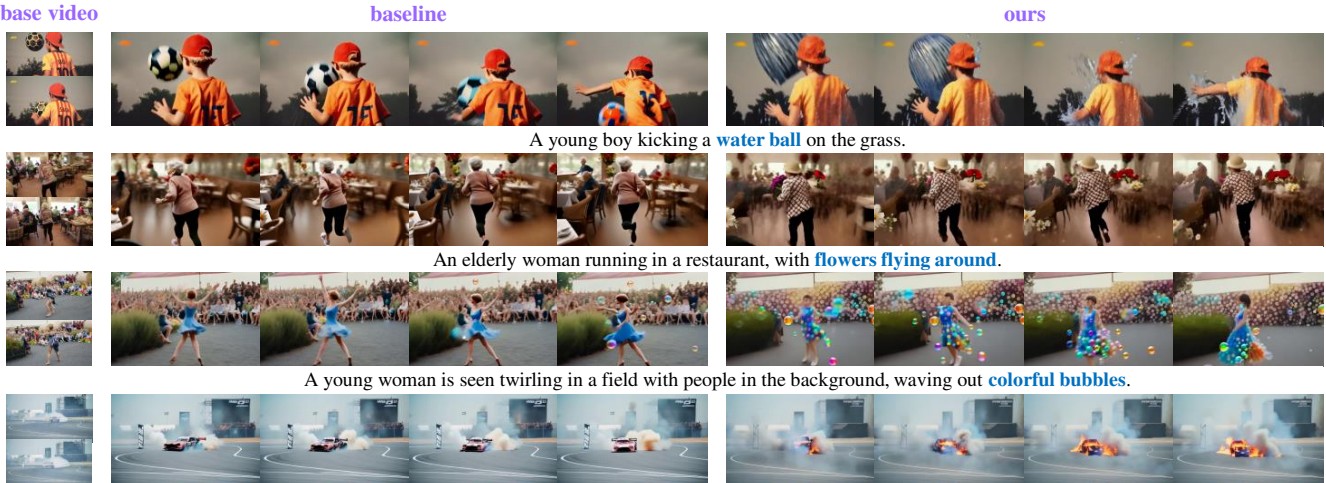

A young boy kicking a **water ball** on the grass.

An elderly woman running in a restaurant, with **flowers flying around**.

A young woman is seen twirling in a field with people in the background, waving out **colorful bubbles**.

Drift car performance on a race track, **surrounded by fires**.

**Figure 6: Qualitative results for synthesis without reference effects (SNRE) of baseline and our improved method.**

to compare. For SRE, we conduct experiments using the pipeline described in Section 4.2 and report qualitative results below. For SNRE, we qualitatively and quantitatively compare the baseline and improved methods described in section 4.3, respectively.

**Metrics.** In the context of synthesizing visual effects without reference, we employ two CLIP scores metrics to evaluate the performance of our method against baseline: **Text:** This metric assesses the alignment between the synthesized video and the effect prompt, measuring how well the video matches the described effects. **Content:** This score determines the similarity between the synthesized video and the base video, aiming to measure how much the output

video modifies the input.

**Synthesis with Reference Effects (SRE).** Figure 5 illustrates the qualitative results of our pipeline for SRE. The base videos represent diverse scenarios where visual effects are to be incorporated. The effect videos demonstrate the specific visual effects used as references, and the output columns display the final synthesized videos. For instance, in the first row, the boxer's fists are enhanced with realistic flames, highlighting our method's ability to adapt effects to various contexts and scales.

**Synthesis without Reference Effects (SNRE).** Figure 6 presents a comparison between the baseline and our improved approach for

**Table 1: Comparison of CLIP scores and User Preferences across different setting. Our method outperforms the baseline in the Text metric. $M_{user}$ and $NP$ indicate our proposed two techniques.**

| Metric | CLIP scores | | User Preferences | | |
|---|---|---|---|---|---|
| | Text↑ | Content | Text | Quality | Temporal |
| w/o $M_{user}$ | 0.1828 | 0.7331 | - | - | - |
| w/o $NP$ | 0.1658 | 0.7874 | - | - | - |
| $\omega$=2.5 | 0.1772 | 0.7786 | - | - | - |
| $\omega$=5.0 | 0.1786 | 0.7648 | - | - | - |
| $\omega$=7.5 | 0.1789 | 0.7487 | - | - | - |
| $\omega$=10.0 | 0.1781 | 0.7376 | - | - | - |
| ours($\omega$=12.0) | **0.1897** | 0.7063 | 68.86% | 34.17% | 45.86% |
| baseline | 0.1786 | 0.7349 | 31.14% | 65.83% | 54.14% |

SNRE. In the first sequence, the baseline method only alters the color of the soccer ball, while our method transforms it into a water ball, complete with a realistic bursting effect that closely matches the textual description. Similarly, in the fourth scene, the baseline only generates smoke around the drifting car, lacking the dynamic fire effect described in the text. Our method, however, successfully synthesizes the flames, adding intensity and excitement to the scene. The second and third rows further highlight the shortcomings of the baseline. It produces minimal floral elements in the restaurant and barely noticeable bubbles in the park, resulting in effects that are not prominent and lack the striking impact required by the scene's description. In the stark contrast, our outputs are rich with blossoms and bubbles, enveloping the subjects in a vibrant and lively manner that greatly aligns with the textual prompts.

The quantitative results are presented in Table 1, which indicate that our method effectively modifies the base video and outperforms the baseline method in generating a video that more closely matches the effect description.

In addition, we conduct a subjective user study to further compare these methods. Specifically, we randomly selected 75 synthesized videos and made a questionnaire, inviting 25 users to vote in terms of text matching, video quality and temporal consistency respectively. The voting results are displayed in Table 1. Our method has a great advantage in text matching degree and is comparable to baseline in temporal consistency. The video quality score measures visual quality without regard to text alignment. Since we use the original caption of the base video as a negative prompt to enhance the expression of visual effects, it inevitably makes the synthesized video lose some details compared to the base video, thus lowering the video quality. In contrast, the baseline scores higher because it tends to ignore visual effects and reconstruct the input.

## 5.2 Ablation study

**Influence of Paste Transparency.** In SRE, we first paste the effects video into the specified position on the base video, and then use the video generator to re-generate the video in order to fuse the two in a more rational way. However, direct pasting would make the effects in the outputs very stiff and abrupt, as shown in subfigure (c) in Figure 7, so we adjust the transparency $\alpha$ of the effect video. That is, the pixels of the base video are linearly combined with the

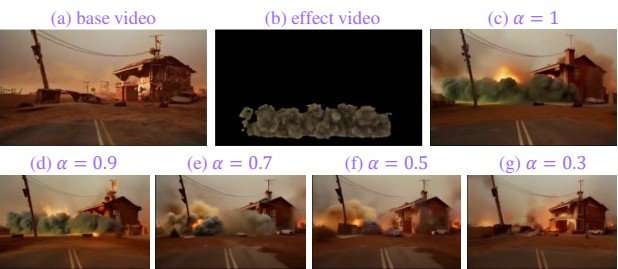

(a) base video (b) effect video (c) $\alpha = 1$
(d) $\alpha = 0.9$ (e) $\alpha = 0.7$ (f) $\alpha = 0.5$ (g) $\alpha = 0.3$

There is an **explosion** in front of a house, ultra realistic.

**Figure 7: Ablation of different transparency $\alpha$ when paste effect video to base video.**

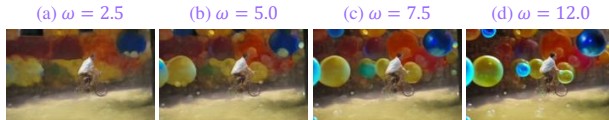

(a) $\omega = 2.5$ (b) $\omega = 5.0$ (c) $\omega = 7.5$ (d) $\omega = 12.0$

A young man is riding his bicycle, with **bubbles appearing surrounding**.

**Figure 8: Ablations of different classifier-free guidance scale of negative prompts.**

pixels of the effect video when pasting by the following formula:

$$P_{init} = (1 - \alpha) * P_{base} + \alpha * P_{ref}, \qquad (4)$$

where $P$ denotes the pixel value at the specified paste position. The experimental results are shown in Figure 7. Lowering the transparency $\alpha$ does result in a better integration of the special effect with the base video, but too low $\alpha$ value will cause the model ignore the presence of the reference effect.

**Influence of Using Original Captions as Negative Prompts.** We employ quantitative metrics to investigate the impact of utilizing the original captions of the base videos as negative prompts in the generation process, as illustrated in Table 1. We use $NP$ in the table to denote this method and calculate metrics for different values of $\omega$. The results shows that implementing original captions as negative prompts improves the alignment between the output video and the effect prompt, and this alignment increases with the escalation of $\omega$. Figure 8 corroborates this trend. Additionally, while this approach amplifies the divergence between the output and the base video, it ensures that the quality remains at an elevated level.

**Influence of Using User Masks as Attention Maps.** First, we explore the impact of different pixel values of non-zero region in the user mask. When the value is 1, the result is shown in subfigure (c) of Figure 9, where the user mask is in the lower left corner. The result indicates that it leads to the model adding an excessive amount of flame effects in the specified area, causing complete distortion of the image in that region. Consequently, we further reduced the pixel values of the non-zero areas in the mask image. The outcomes of this adjustment are displayed in Figure 9. In conclusion, we chose 0.3 as the pixel value for the non-zero region.

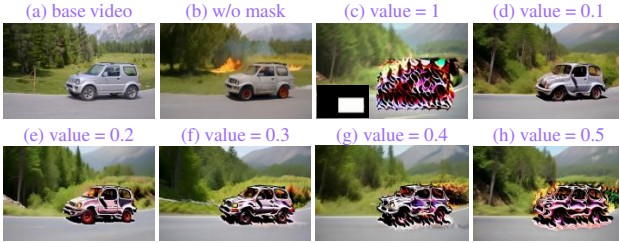

A gray car tightly **surrounded by flames** is driving normally on a mountain street.

**Figure 9: Ablations of different value of non-zero region in user masks.**

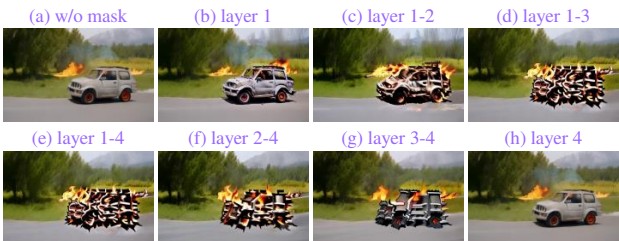

A gray car tightly **surrounded by flames** is driving normally on a mountain street.

**Figure 10: Ablation of different action position of the user masks in the UNet layers.**

Furthermore, we experimentally investigate the impact of using user masks at different positions within the UNet. Specifically, we divide the UNet into four layers from top to bottom, from layer 1 to layer 4, according to the feature dimensions in descending order within the UNet architecture. We replace the attention maps with the user mask at various positions within these four layers, and the results are illustrated in Figure 10. As can be seen, the use of masks in the bottom layers is not as effective as in the top layers, especially when replacing only in the fourth layer, the replacement of the attention map is almost ineffective. We finally chose to use the user mask in layers1-2.

## 6 Discussions

Our experiments have demonstrated the feasibility and potential of integrating AIGC with visual effects synthesis. In this section, we will discuss some issues identified during our research that warrant further extensive consideration.

**Scale of Dataset.** In this paper, we compiled an effects dataset from the internet, covering a wide range of effect categories commonly seen in the film and television industry. However, limited by our lack of professional experience in effects production, we could not produce or collect a more extensive array of effects. This prevents our dataset from supporting training or large-scale validation.

Moreover, for SNRE, our data acquisition method involved manually captioning base videos and modifying these to include descriptions of effects. This process also restricted the scale of our dataset.

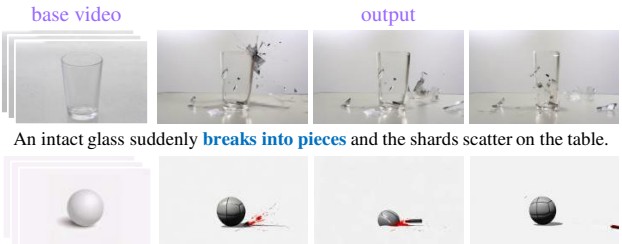

An intact glass suddenly **breaks into pieces** and the shards scatter on the table.

A bullet crosses the ball and **breaks it**.

**Figure 11: Bad cases.**

Perhaps, we could start by creating a series of brief descriptions of individual effects, and then leverage the capabilities of large language models to automatically create descriptions containing effects for the base videos, in conjunction with the existing video-caption dataset. This would generate a much larger dataset that would provide the basis for model training.

**Deficiencies in Rigid-body Generation.** Visual effects commonly used can broadly be categorized into four types: particle effects, fluid effects, rigid-body effects, and soft-body effects. Particle effects refer to those produced by simulating the movement of a multitude of particles, such as explosions, smoke, sparks, and meteors and so on. Fluid effects are primarily utilized to simulate the flow of liquids, such as oceans and waterfalls. Rigid-body effects simulate the movement of hard objects within real-world environments, examples include a wrecking ball hitting a wall or glass shattering. Conversely, soft-body effects model the behaviors of various elastic objects in the real world, such as jello or balloons.

Our experiments demonstrated the competitive potential of AIGC technology in generating fluid and particle effects. However, we also observed that the model lacked experience in generating soft-body and rigid-body effects, particularly with rigid-body effects. As shown in Figure 11, even in simple scenarios, our method fails to achieve effects such as "break". This limitation might be attributed to the capabilities of our backbone model Videocrafter2. Indeed, we noticed that even the most impressive text-to-video model to date, Sora, struggles with understanding the concept of a "cup shattering". This suggests that generating rigid bodies is not only a challenge for AIGC-based visual effects synthesis but also a broader issue faced by the pursuit of a "world simulator" in current research.

## 7 Conclusion

In this work, we explore the integration of visual effects synthesis with the burgeoning field of AIGC. Our experimental findings demonstrate the substantial potential of this integration, where pre-trained video generator can seamlessly blend effects with scenes and adjust scene elements logically. We believe that advancing research in AIGC-based visual effects synthesis will transform film and television production by enabling streamlined processes and innovative applications, while also allowing non-professional users to create high-quality visual effects videos, opening new creative opportunities for a broader audience.

## 8  Acknowledgments

This study is supported by grants from the National Natural Science Foundation of China (Grant No. 62122018, No. 62020106008, No. U22A2097, No. U23A20315), and Kuaishou.

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
