# OpenReview forum: "MagicVFX: Visual Effects Synthesis in Just Minutes"
_acmmm.org/ACMMM/2024/Conference — MM2024 Poster_

### Official Review · Reviewer_5jki · 2024-05-16

**Rating:** 5
**Confidence:** 3

**Summary:**

The authors compile an effects dataset from the internet, encompassing a wide range of effect categories. Furthermore, they propose an approach for synthesising visual effects for two paradigms - with reference effects and without reference effects.

**Strengths:**

- Paper is well written and structured, and the figures are very useful to understand the approach.
- Authors have compiled a novel dataset for visual effects in videos, which is beneficial for further research in this field.
- Proposed pipeline for visual effects synthesis is straightforward and sound.
- Good evaluation using qualitative scores (human preferences from a user study) .
- Ablation study is well done.

**Limitations:**

- The user study is done only with 9 users. That's quite a small sample for a user study, at least twice as much users would be good.
- Method performs worse in video quality than the baseline.

**Suitability:**

3

---

### Official Review · Reviewer_9hju · 2024-05-25

**Rating:** 5
**Confidence:** 3

**Summary:**

This paper presents a visual effects synthesis study. A dataset and a model are proposed. Specifically, the proposed method can achieve two effects, including visual special effects synthesis with reference effects and visual special effects synthesis with texts.

**Strengths:**

1. The results are impressive, especially for the text-controlled effect generation.
2. The method supports both video effect integration and text-guided integration.
3. The paper is well written and well organized.

**Limitations:**

1. In Fig. 4, it is clear how to add video effect to the base videos. However, the method of adding effect using text is not clear.
2. The pipeline of text editing is not clear. Can we add the effect to anywhere described in the text? How to realize it.
3. How to train the whole framework, since the ground truth seems unknow.
4. The references are too few, seem insufficient

**Suitability:**

3

---

### Official Review · Reviewer_kTz6 · 2024-05-28

**Rating:** 3
**Confidence:** 3

**Summary:**

This paper proposes MagicVFX, a new framework to integrate generative AI, specifically text-to-video models, into the VFX production pipeline. The goal is to increase controllability and reduce manual costs in VFX systems. To address this, the paper introduces a new dataset consisting of reference-based and reference-free visual effect videos with captions. The authors also present a pipeline that uses a text-to-video diffusion model to integrate visual effects (either with or without reference effects) into a base video. Using the proposed dataset and pipeline, the authors provide experimental results demonstrating the effectiveness of MagicVFX in successfully adding visual effects to the original videos.

**Strengths:**

* The core problem addressed in the paper is both new and potentially has significant implications for the VFX industry.

* The proposed dataset can be used for future evaluation of AIGC-based VFX systems.

* The proposed method is simple to implement, making it accessible to a broader audience.

* The depth of the experimental results and ablations is reasonable, covering the important aspects and design choices of the method.

* The writing of the paper is clear and easy to read for the most part.

**Limitations:**

- The comparison with current video editing systems is missing. I acknowledge that the authors briefly mention this in the related work section and Figure 2, but I am not yet convinced why existing video editing systems cannot tackle this task. If the authors could provide more experimental results on this, as well as detail the limitations of current video editing systems compared to the proposed method with visual examples (e.g., cases where using existing editing methods fail but the proposed method succeeds in adding the visual effect to the base video), it would significantly enhance the overall contribution of the paper.

- The authors use negative prompting to enhance the quality of visual effects, but since they use the base caption as the negative part, it reduces the quality of the generated video compared to the baseline. I am curious to see an example where two CFG signals are used (with the null prompt for the negative part): one that uses the base caption and one that uses the visual effect part of the caption. i.e., using $ \hat{\epsilon} = \epsilon(x_t, t, c_{base}) + w_1 (\epsilon(x_t, t, c_{base}) - \epsilon(x_t, t, c_{null})) +  w_2 (\epsilon(x_t, t, c_{VFX}) - \epsilon(x_t, t, c_{null})).$ It would be interesting to see if this modification results in both high-quality generation and visual effects with proper $w_1$ and $w_2$.

#### **Minor Comments**

- Please provide more implementation details regarding the noise level, sampler, and details of the attention-based editing in the appendix for better reproducibility of the results.

- As similar ideas have been used in other image and video editing papers, please provide a more detailed analysis of the related work to further distinguish the novelty of the method. For instance, you should cite the SDEdit [1] paper for the idea of adding noise and denoising, and also compare to [2, 3].

[1] Meng, Chenlin, et al. "Sdedit: Image synthesis and editing with stochastic differential equations." arXiv preprint arXiv:2108.01073 (2021).

[2] Esser, Patrick, et al. "Structure and content-guided video synthesis with diffusion models." Proceedings of the IEEE/CVF International Conference on Computer Vision. 2023.

[3] Wang, Xiang, et al. "Videocomposer: Compositional video synthesis with motion controllability." Advances in Neural Information Processing Systems 36 (2024).

#### **Final Thought**

I will be happy to adjust my score if the authors provide more explanation on the limitations during the rebuttal.

**Suitability:**

3

---

### Official Review · Reviewer_vX5e · 2024-05-31

**Rating:** 2
**Confidence:** 3

**Summary:**

The paper introduces AIGC techniques for visual effect synthesis, which generate complex, high-quality special effects in videos in only a few minutes. Moreover, an experiment was conducted to demonstrate that the pipeline proposed by this paper could produce impressive visual effects synthesis outcomes.

**Strengths:**

The paper collects a basic dataset encompassing common visual effects for assessing the ability of visual effects synthesis methods. It builds a training-free base model with extensive experiments and demonstrates the potential of the pre-trained video diffusion model in visual effect synthesis. The paper, in general, looks into a relevant problem for MM.

**Limitations:**

1. This paper implements the task of visual effects synthesis using a pre-trained text-to-video generation model. However, the model’s name (VideoCrafter2) is only mentioned in Section 1&2. It is unclear why the authors selected VideoCrafter2 as their backbone model. Is there any reason why VideoCrafter2 is better than other pre-trained text-to-video generation model?

Besides, the method this paper proposed is somewhat straightforward. The pipeline for synthesizing visual effects seems to mostly rely on the backbone pre-trained model, which is a previous contribution of another work. Therefore, this paper's contribution seems not outstanding.

2. This paper claims that it is the first work to explore AIGC-based visual effects synthesis, and there is no applicable methodology to compare. Please verify this argument carefully. If it is the first work, it is more convincing to transfer other similar works to this field and compare this method with them. Besides, since this paper uses VideoCrafter2 as the pre-trained model, some other pre-trained models can be chosen as baselines to compare with (Page 5, Section 5.1).

3. This paper invites nine users to conduct a user study and compare different methods. The number of users is relatively small, which diminishes the study's contribution.

4. A conclusion section seems necessary for this paper. However, the authors combined it with the discussion, making the paper less easy to read.

5. According to the results of the subjective user study, the method proposed in this paper has a much lower video quality score compared to the baseline. Although the authors explained it as the synthesized video losing some details, is this the only reason? More explanations and discussions are needed.

6. Considering the reproducibility of this study, please add the questionnaires used in the subjective user study as support material or appendix.

Minor comments
Page 7, Section 5.1, Line 720. There should be a space between the period and "Our".
Page 5, Section 4.3, Line 544. The “predict” should be “predicts”.
Page 8, Section 6, Line 899. “visual effects” should be “Visual effects”.

**Suitability:**

3

---

### Meta-Review · Area_Chair_uQFq · 2024-07-06

**Recommendation:** Accept (Poster)
**Confidence:** 4

**Metareview:**

The paper introduces AIGC techniques for visual effects, which generate complex special effects in videos in just a few minutes.  For that, the paper proposes a new dataset and a pipeline that uses text-to-video diffusion.  The method can generate effects with or without reference.  The core problem the paper addresses is very interesting and has a potential broad impact on the VFX industry.  The proposed method is technically sound, and the presented results are very interesting.  The reviewers raised concerns about the initial state of the paper, e.g., the user study being done with only 9 users, the lack some implementation details, and asked the authors for additional details regarding the subjective study.  During the rebuttal phase, the authors extended the study to include 16 additional participants and answered most of the reviewers' concerns.

As final scores, three reviewers recommend the acceptance (1 Accept and 2 Weak Accept), while one reviewer rates it as borderline reject.  Given the importance of the topic, the exciting results, and the fact that the proposed dataset can contribute to advancing similar research in the area, the recommendation is to accept the paper.